# The Prognostic Value of Liquid Biopsies for Benefit of Salvage Radiotherapy in Relapsed Oligometastatic Prostate Cancer

**DOI:** 10.3390/cancers14174095

**Published:** 2022-08-24

**Authors:** Natalia V. Bogdanova, Hoda Radmanesh, Dhanya Ramachandran, Anne Caroline Knoechelmann, Hans Christiansen, Thorsten Derlin, Christoph Alexander Joachim von Klot, Roland Merten, Christoph Henkenberens

**Affiliations:** 1Department of Radiation Oncology, Hannover Medical School, 30625 Hannover, Germany; 2Gynecology Research Unit, Clinics of Obstetrics and Gynaecology, Hannover Medical School, 30625 Hannover, Germany; 3Department of Nuclear Medicine, Hannover Medical School, 30625 Hannover, Germany; 4Clinic for Urology, Hannover Medical School, 30625 Hannover, Germany

**Keywords:** liquid biopsies, circulating tumor cells, oligometastatic prostate cancer, PSMA-PET, salvage radiotherapy

## Abstract

**Simple Summary:**

Around 30% of patients with oligometastatic prostate cancer relapse will benefit from local PET/CT-guided ablative radiotherapy (RT) with improved progression-free and ADT (Androgene Deprivation Therapy)-free survivals. Therefore, there is an urgent need for predictive testing for therapeutic benefits prior to initiation. Various tests have already been established on tumor specimens for the prediction of prostate cancer’s behavior or therapy outcome. However, in imaging-proven relapse tumor tissue from the local recurrence or metastases is often not available. Hence, there is a need for a liquid biopsy-based testing. We aimed to assess the prognostic value of CTCs- associated mRNA and blood-derived RNA for the benefit of PSMA PET-guided salvage RT in oligometastatic prostate cancer relapses. Significant correlations were found between the relative transcript levels of several investigated genes and clinicopathological parameters. Furthermore, distinct “transcriptional signatures” were found in patients with temporary and long-term benefits from RT.

**Abstract:**

To assess the prognostic value of “liquid biopsies” for the benefit of salvage RT in oligometastatic prostate cancer relapse, we enrolled 44 patients in the study between the years 2016 and 2020. All the patients were diagnosed as having an oligometastatic prostate cancer relapse on prostate-specific membrane antigen (PSMA)-targeted PET-CT and underwent irradiation at the Department of Radiotherapy at the Hannover Medical School. Tumor cells and total RNA, enriched from the liquid biopsies of patients, were processed for the subsequent quantification analysis of relative transcript levels in real-time PCR. In total, 54 gene transcripts known or suggested to be associated with prostate cancer or treatment outcome were prioritized for analysis. We found significant correlations between the relative transcript levels of several investigated genes and the Gleason score, PSA (prostate-specific antigen) value, or UICC stage (tumor node metastasis -TNM classification of malignant tumors from **U**nion for **I**nternational **C**ancer **C**ontrol). Furthermore, a significant association of *MTCO2*, *FOXM1*, *SREBF1*, *HOXB7*, *FDXR,* and *MTRNR* transcript profiles was found with a temporary and/or long-term benefit from RT. Further studies on larger patients cohorts are necessary to prove our preliminary findings for establishing liquid biopsy tests as a predictive examination method prior to salvage RT.

## 1. Introduction

For the evaluation of the biological behavior and prognosis of a prostate carcinoma, the Gleason score [1], the Risk Group according to Epstein [2], and the tumor node metastasis (TNM) stage [3] have proven for decades. In recent years, molecular-genetic testing of tumor material has been shown to be an additional independent predictor [4,5]. A relapse after initial curative therapy of prostate cancer affects a significant number of patients and can be indicated biochemically by an increase in prostate-specific antigen (PSA). The level of the PSA value usually correlates with the tumor load. When later metastases or a local recurrence occur, in many cases, a genetically different tumor from the one at primary diagnosis is present [6]. While larger metastases would be accessible for a renewed biopsy, in the recurrence that is initially detected only biochemically, very small metastases are often present, which are practically inaccessible for biopsy. The localization of new tumor foci is very problematic if the PSA values are low. With the clinical introduction of prostate-specific membrane antigen ligand positron emission tomography (PSMA-ligand PET) it became possible to identify recurrences with a high diagnostic accuracy at an early stage and at low PSA values [7]. This is particularly true for lymph node recurrences [8]. In such a clinically common situation, interdisciplinary monitoring is initially carried out by durable palliative Androgene Deprivation Therapy (ADT), which is still the guideline, with or without adding docetaxel or bisphosphonates [9,10,11,12,13]. However, in this mostly oligometastatic situation, local PET/CT-guided ablative radiotherapy (RT) may represent a treatment option for some third of the patients that will profit with improved progression-free and ADT−free survival, although an RT benefit has not yet been demonstrated in prospective randomized trials. Therefore, there is an urgent need for predictive testing for a therapeutic benefit prior to the initiation of the RT of a recurrence detected in PSMA-PET. To date, no clear parameters have been identified to predict the outcome of PSMA-guided RT for oligorecurrence. Despite the defined clinical and imaging inclusion and exclusion criteria, the investigated collectives of the pre-PSMA-PET era are biologically so heterogeneous that it cannot be precisely predicted which patient will benefit greatly, which will benefit in the short to medium term, and which patients will not benefit. The most likely explanation is that current therapy concepts are based almost exclusively on the quantity of the recurrence (number of lesions). Molecular tumor biology has not yet been adequately correlated with modern imaging and the clinical course (including the therapeutic response) prospectively, since reliable molecular markers are missing in this oncological setting. Therefore, molecular hematogenous biomarkers are required that, in addition to the isolated local recurrence and the extensive polytopic metastasis, can measure an intermediate tumor stage between these “oncological poles” [14,15]. Ideally, such markers should be used with limited aggressiveness and metastatic capacity and should provide the possibility of evaluating the effectiveness of the therapy. The progression of the PSA value alone is not sufficient here. Although it reliably indicates both the failure after primary therapy and the further progress after relapse therapy, it cannot adequately predict which patients will benefit to what extent. PSA levels trigger diagnosis and treatment, but their use for predicting therapy outcomes is limited. Therefore, there is a need for biomarkers that are highly correlated with the clinical course and diagnostic imaging while measuring the aggression of the tumor disease as well as the therapy outcome. While a biopsy of the metastasis is often unavailable, the use of a “liquid biopsy”—analyzing blood-derived nucleic acids, which provide an overview of patients’ genetic background, or tumor cells, and tumor cell products in blood samples—might overcome these difficulties even for multiple metastases [16]. Thus, molecular characterization should not only quantitatively measure the number of circulating tumor cells (CTCs) in blood but also qualitatively characterize the “landscape” of these cells, for instance, by the expression of PSMA/PSA on the CTCs’ surface and tumor cell-associated mRNA. The intrinsic germline features of patients also play a very crucial role here. We aimed to assess the prognostic value of “liquid biopsies” (CTC-associated mRNA and blood-derived RNA) for the benefit of PSMA PET-guided salvage RT in oligometastatic prostate cancer relapse.

## 2. Materials and Methods

### 2.1. Study Design and Patients Selection Criteria

The protocol was reviewed and approved by the Hannover Medical School Institutional Ethics Committee Board (ethic vote no. 8265_BO_K_2021). Informed written consent was obtained from each patient prior to sample collection. Patients were informed that the results of the study may be published in scientific journals and presented at medical meetings and can be made available for other researchers to use in other research projects, but that the information that directly identifies any person would not be used.

All enrolled patients had prostatectomy or completed definitive RT (76 Gy in 38 Fractions or 60 Gy in 20 Fractions) with a curative intent for a primarily regionally localized prostate carcinoma and were admitted to the Department of Radiation Oncology at Hannover Medical School for follow-up with increasing PSA levels between the years 2016 and 2020. Patients underwent PSMA-targeted PET-CT and were diagnosed as having oligometastatic disease. Patients were selected for salvage RT with a maximum number of 4 PSMA-positive lesions, including local recurrence, pelvic lymph node metastasis, or distant metastasis. All detected lesions were irradiated. Various doses and fractionations were used as part of routine clinical treatment. Forty-four patients gave informed consent for a single sample of 15 mL of venous blood before start of RT. Blood sampling was successful for 43 patients. Median follow-up of 14.8 months was available for all patients. Upon acquiring clinical data for follow-up, we found that 13 patients additionally received ADT concurrent to RT; thus, the efficacy of RT could not be determined with certainty for these patients. Thus, statistical analysis of RT benefit was also performed with the exclusion of these patients. Acute and chronic toxicity follow-ups with PSA-determination every three months and questioning of further disease progression were performed as part of routine clinical care. The initial PSA value at the time of first diagnosis (iPSA) was 12.15 ng/mL (range 2.6–40.43 ng/mL). The PSA doubling time after primary treatment until recurrence was 22 months on average (range 2–61 months). PSA at the time of PSMA-PET-CT was 2.2 ng/mL on average (range 0.09–64 ng/mL). Gleason sum averaged 7.6 (range 6–9). T-stage was on average 2.5 (range 1–4). A total of 5/43 (11.6%) of patients had metastases to regional lymph nodes at primary therapy (N1), whereas 38/43 (88.4%) had none (N0). A total of 13/43 (30%) had no remission after RT of all PET-detected recurrences, 25/43 (58%) had remission that remained stable for 4–12 months, and 5/43 (12%) had remission that lasted longer than 12 months. The time to progression (TTP) was calculated from the start of RT until PSA increased or imaging evidence of new metastases occurred. For this cohort, TTP was 7.1 months on average with a median TTP of 6 months (range 3–19 months). Clinical characteristics of the patients are summarized in Table 1.

### 2.2. Gene Panel Development

We reviewed published data for gene expression profiles or gene mutations corresponding to prostate cancer. Genes known or suggested to be associated with disease or treatment outcome were browsed on NCBI (including GEO—Gene Expression Omnibus) (https://www.ncbi.nlm.nih.gov/geo/), COSMIC (https://cancer.sanger.ac.uk/cosmic), TCGA (The Cancer Genome Atlas) (https://www.cancer.gov/about-nci/organization/ccg/research/structural-genomics/tcga), and GENT (Gene Expression database of Normal and Tumor tissues, http://gent2.appex.kr/gent2/) databases, status as of march 2020.

Using QIAGEN Ingenuity Pathway Analysis and STRING database, the best candidates from the most representative (highly significant) signaling pathways for the test panel were prioritized. In total, 54 candidates were selected for our study (Figure 1, lower panel).

### 2.3. Samples Collection and Processing

Since liquid biopsies are relatively stable in EDTA tubes for up to 12 h before processing [17] and since our study is monocentric at this stage with no need for long storage, whole blood samples for each patient were collected in two 10-mL EDTA monovettes and immediately handled. Whole blood from one 10-mL EDTA monovette was used for isolation of CTCs (Figure 1, upper panel). Two commercial kits were employed in parallel: AdnaTest Prostate Cancer Select/Detect (Qiagen, Hilden, The Netherlands), which recognizes *PSMA*, *PSA,* and *EGFR* (with *beta-Actin* included as a control), and Dynabeads™ Epithelial Enrich Kit (EpCAM+, Thermo Fisher Scientific, Waltham, MA, USA). CTCs, isolated by Dynabeads™, were processed by Dynabeads™ mRNA DIRECT™ Purification Kit. CTCs, isolated by the AdnaTest Select, were processed by AdnaTest Prostate Cancer Detect following the manufacturer’s instructions, with minimal changes: the initial washing step was performed with PBS + 0.05% bovine serum albumin, which reduced the loss of magnetic beads to plastic surfaces [18]. After direct amplification (AdnaTest), the samples were loaded on a 2% agarose gel and semi-quantified. The method used is based on ImageJ gel analysis (ImageJ v.1.8.0), which routinely quantifies the density of the bands (the ImageJ users guide can be viewed at: https://imagej.nih.gov/ij/docs/guide/index.html, protocol used was modified from following tutorial: https://www.unige.ch/medecine/bioimaging/files/2014/1208/6025/GelAnalysis.pdf, and [19]). Our protocol enables the relative quantification of PCR product bands of interest (*PSMA*, *PSA,* and *EGFR*) normalized to the loading-control band (*ACTB*) as well as to the standard sample (positive control, provided by the manufacturer). The final quantification reflects the relative amounts as fold-changes in relation to the standard. The manufacturer defined the presence of CTCs as any one of *PSMA*, *PSA,* or *EGFR* expressions.

PSA positivity is assumed as successful detection of CTCs, as described in [20,21]. Thus, upon isolation by Dynabeads™ and cDNA synthesis by SuperScript™ IV VILO™ (Thermo Fisher Scientific), CTCs were identified through *cytokeratin 19* (which belongs to the key components of intermediate fibers of the cytoskeleton in epithelial cells and is expressed in normal epithelial cells, epithelial tumors, and metastatic cells [22]) and *KLK-3*-*PSA* positivity in SYBR-Green qRT-PCR (quantitative Real-time PCR). Further, qRT-PCR analysis (TaqMan method) of androgen receptor (*AR*) and *AR* splice variants (*AR.V3*, *AR.V7*, *AR.V9,* and *AR.V12- AR.V567es*) was performed.

From the second 10-mL EDTA monovette, 3 mL of whole blood was aliquoted in 0.5 mL volumes and stored at −80 °C. The rest of the blood was used for plasma isolation through Ficoll density gradient. Plasma was aliquoted in 0.5 mL volumes and stored at −80 °C. Any remaining blood or plasma aliquots were submitted to the internal Biobank of Hannover Medical School for long-term storage. For isolation of total RNA from blood specimens, RiboPure™ RNA Purification Kit (ThermoScientific) was used according to the manufacturer’s instructions. Isolated biological samples were processed with SuperScript™ IV VILO™ for cDNA synthesis and the generated cDNAs were utilized for high-throughput microfluidic qRT-PCR on the Biomark HD platform (Fluidigm, South San Francisco, CA, USA) and TaqMan or SYBR-Green reactions. 

### 2.4. Quantitative Real-Time PCR (qRT-PCR)

We performed pre-amplification of all transcripts in order to increase the available amount of cDNA template per reaction and to enhance the detection of the target transcripts for qRT-PCR analysis via SYBR-Green, TaqMan, or microfluidic techniques. All experiments included four measurements (quadruplicates) for each sample and negative cDNA-control, as well as water controls. A total of up to 40 ng of RNA was used for cDNA synthesis followed by 10 cycles of pre-amplification reaction. SsoAdvanced™ PreAmp Supermix (BioRad, Hercules, CA, USA) was used (according to the manufacturer’s instructions) with a pool containing all primers investigated by SYBR-Green and TaqMan methods, at a final concentration of 500 nM each.

SYBR-Green qPCR reactions were performed using appropriate PCR assays along with the SsoAdvanced Universal SYBR Green Supermix (BioRad) and the following PCR protocol: 95 °C × 10 min, 95 °C × 15 s, and 60 °C × 60 s for 45 cycles, followed by melting curve analysis.

TaqMan qPCR reactions were performed using appropriate PCR assays along with the TaqMan Fast Advanced Master Mix (ThermoFisher Scientific, Waltham, MA, USA) using the following protocol: 95 °C × 10 min, 95 °C × 15 s, and 60 °C × 60 s for 45 cycles. 

SYBR-Green and TaqMan approaches were carried out on the CFX384 Touch Real-Time PCR Detection System (BioRad), with a reaction volume of 10 ul including 1.5 ul cDNA (after pre-amplification and two-fold dilution) per reaction.

In a nanofluidic automated real-time PCR system (Fluidigm), based on microfluidic technology, dynamic arrays of integrated fluidic circuits (IFCs) were used. IFCs possess special controlled valves and interconnected channels, which enables an automatic mix of biological samples and reagents in a controlled manner. We used a dynamic array for qPCR, in which the chip format allows for 9216 PCR reactions (96.96 chip format; 96 samples × 96 assays) in a single qPCR run (https://www.fluidigm.com/),). This cost-effective and time-saving technique reduces the reaction volume to 10 nL and requires 10 cycles of sample pre-amplification, which were performed using the Preamp Master Mix (Fluidigm) and a pool of 34 Delta Gene™ assays (Fluidigm) representing all investigated gene transcripts (except housekeepers) at a final concentration of 500 nM each. Pre-amplified cDNA samples were cleaned with an exonuclease I reaction (New England Biolabs, Ipswich, MA, USA) and were diluted two-fold prior to loading onto the chip, as per manufacturer’s recommendations (PN 68000088 N1). The qRT-PCR run was performed for 45 cycles as per manufacturer’s instructions (Protocol name: GE Fast 96 × 96 PCR + Melt v2.pcl). *ACTB*, *B2M*, *GAPDH HPRT1, MRPL19, RPL13A,* and *RPLP0* gene transcripts were used as housekeepers. All used primers/probes are available on request.

### 2.5. Data Processing and Statistical Evaluations

A gene was noted to be expressed when at least three replicates of each sample (pipetted in quadruplicates) revealed detectable Ct (cycle threshold) values (≤40). Obtained qPCR results were analyzed in QBase+ software (version 3.3, Biogazelle, Gent, Belgium), whose algorithms enable the processing of data according to international guidelines (https://www.qbaseplus.com/). QBase+ software utilizes delta Ct method for the normalization to the reference genes (housekeepers) and enables the performance of this mathematical operation for multiple housekeeper genes at once followed by a calculation of the amplification efficiencies. Results were exported after logarithmic-transformation, including standard errors, and were utilized for further statistical analysis.

We used GraphPad Prism Software for statistical calculations (version 9.4.0; GraphPad, San Diego, CA, USA). In order to compare differences between two groups, a student’s *t*-test was performed. For multiple comparisons between consecutive groups, an ANOVA was performed followed by a linear trend test. As indicated, regression analysis or chi square test (OpenEpi—2 × 2 Table Statistics, source: Dean AG, Sullivan KM, Soe MM. OpenEpi: Open Source Epidemiologic Statistics for Public Health, Version. www.OpenEpi.com, updated 2013/04/06) was performed for assessing the correlations between clinico-pathological parameters. Pearson’s correlation analysis for RT benefit was performed with a *p* value < 0.05 defined as statistically significant. For evaluation of therapeutic benefit, patients with decreased or stable PSA-levels in serum and no progress in imaging longer than 12 months were defined as high responders (benefit level 2). Patients with decreased or stable PSA-levels and no progress in imaging shorter than 12 months were defined as low responders (benefit level 1). All patients with increased PSA-levels in early follow up or with progress in imaging were defined as non-responders (benefit level 0).

## 3. Results

### 3.1. Transcript Expression of Patient-Derived CTCs—Associated mRNA

In total, 43 samples were analyzed (Figure 2). CTCs were detected in 42 samples (97.7%), as those were positive for *PSA*. No *EGFR* was detected, and 7/42 (16.7%) samples were positive for *PSMA*. Quantitative RT-PCR showed that *AR-FL* was expressed in 17 out of 42 samples (40.5%), *AR.V7* was expressed in 33 samples (78.6%), and *AR.V567es* (three independent primer pairs: *AR.V12*, *AR.V567-1,* and *AR.V567-2*) was expressed in 38–40 samples (88.1–95.2%). The variability in the estimation of *AR-V567es* that we found has been described previously [20]. In addition, 39 out of 42 samples expressed *AR.V3* (92.9%) and 41 (97.6%) samples expressed *AR.V9*. At least one ARV was found in all CTC-positive samples and 40 samples (95.2%) expressed both *AR.V7* and *AR.V567es*, which are considered the most relevant variants in the ARV-dependent mechanisms of resistance to ADT [23]. 

A known co-expression of AR splice variants (e.g., *AR.V3*, *AR.V7,* and *AR.V9*) was noticeable in our settings and we performed correlation analyses for all the investigated ARVs (Figure 3, upper panel). Since 12 patients in our cohort received ADT concurrently to RT, we performed analyses not only on the whole cohort but also within two groups, patients with and without ADT (ADT+ and ADT−, respectively), and observed significant correlations with the clinical variables (Figure 3, lower panel). 

The *PSA* transcript levels were found to correlate with the PSA values at PSMA-PET in the whole cohort (Pearson’s R = −0.33, *p* = 0.03), iPSA value in the ADT− patients (Pearson’s R = −0.64, *p* = 0.0004), and the number of active loci (Pearson’s R = −0.58, *p* = 0.04) or Gleason score at initial diagnosis (Pearson’s R = 0.56, *p* = 0.04) in the ADT+ patients. The PSA values at PSMA-PET were correlated with PSMA-positive lesions (number of active loci) in the whole cohort (Pearson’s R = 0.51, *p* = 0.0005), in the ADT− patients (Pearson’s R = 0.51, *p* = 0.004) and ADT+ patients (Pearson’s R = 0.56, *p* = 0.05), and with peak uptake in the whole cohort (Pearson’s R = 0.84, *p* = 1.8 × 10^−11^) or in the ADT+ patients (Pearson’s R = 0.86, *p* = 0.0004). Further, the peak uptake in the whole cohort was found to correlate with the number of PSMA-positive lesions (Pearson’s R = 0.40, *p* = 0.01); the Gleason score at initial diagnosis was correlated with tumor relapse (Pearson’s R = 0.38, *p* = 0.05) in the ADT− cohort or with the number of positive lesions in the ADT+ patients (Pearson’s R = −0.56, *p* = 0.05). The known positive association between *AR.V7* expression and ADT treatment did not manifest in our cohort, since *AR.V7* expression was not associated with a history of treatment, but it was found to correlate with the PSA doubling time in the patients who received ADT (Figure 3, ADT+ Group, lower panel, Pearson’s R = 0.68, *p* = 0.02). *AR.V576es* was found to correlate overall (all patients included) with the TNM score (Pearson’s R = 0.33, *p* = 0.05) and *PSA* transcript expression in the CTCs (Pearson’s R = 0.39, *p* = 0.001). Further, we observed a highly significant association of *TMPRSS2* transcript expression, which is commonly used as a readout for androgen activation, with the use of ADT (Mantel–Haenszel chi square 2-tailed test *p*-value = 0.008). *TMPRSS2* transcript expression was additionally found to correlate with the Gleason score at initial diagnosis (Figure 3, Pearson’s R = 0.82, *p* = 0.007) and *AR.V7* transcript expression (Pearson’s R = −0.67, *p* = 0.05) overall. Further, *AR.V9* transcript expression was found to correlate with *PSA* transcript expression in the whole cohort (Pearson’s R = 0.43 *p* = 0.005) and in the ADT− group (Pearson’s R = 0.43, *p* = 0.03), as well as with the Gleason score in the ADT+ group (Pearson’s R = 0.60, *p* = 0.03). *AR-FL* transcript expression was found to correlate with *PSMA* transcript expression in the ADT− patients (Pearson’s R = −0.74, *p* = 0.03). Although the expression of *cytokeratin 19* should just represent tumor cells as cell type-specific epithelial markers, we found correlations with Gleason scores in the whole cohort (Pearson’s R = −0.35, *p* = 0.02) and with the TNM score in the ADT+ group (Pearson’s R = −0.78, *p* = 0.008). The only parameters observed to correlate with a RT benefit were the number of PSMA-positive lesions (Pearson’s R = −0.39, Pearson *p* = 0.01) in the whole investigated cohort and PSA value at PSMA-PET in the whole investigated cohort (Pearson’s R = −0.39, *p* = 0.009) and in the ADT− cohort (Pearson’s R = −0.49, *p* = 0.006). We found no significant difference for RT Benefit between ADT+ and ADT− patients (Mantel–Haenszel chi square 2-tailed test *p*-value = 0.37).

### 3.2. Transcript Expression in Blood-Derived mRNA

We investigated the expression of our pre-selected transcript panel (Figure 1, lower panel, set 2) in the total RNA from whole blood specimens in the investigated patient cohort. The expression levels of the analyzed transcripts were very heterogeneous, partly without a signal for some patients’ RNA samples, although the housekeeper gene transcripts revealed detectable Ct values. We found some significant associations: nine transcripts (*AEN*, *ATM*, *CDH1*, *HOXB7*, *IRF2*, *MT-RNR1*, *MYC*, *NFE2L2*, and *TMPRSS2*) were associated with the Gleason score at the initial diagnosis after a linear trend test (Figure A1), two (*CD4* and *CDKN1A*) with the TNM score (Figure A2A), and one (*IRF2*) with disease recurrence (Figure A2B). A regression analysis revealed that *MTP-ATP6* transcript levels correlate with iPSA value, *KIF20A* with peak uptake, and *SIM2* with PSA doubling time (Figure A2C). The analysis for RT benefit disclosed that seven transcripts (*MTCO2*, *ELK1*, *IRF2*, *MAPK4K*, *NFE2L2*, *ATM*, and *CDC25*) were associated with disease remission (Figure 4). In the linear trend analysis, except for *MTCO2*, four other transcripts (*FOXM1*, *SREBF1*, *HOXB,* and *FDXR*) showed a correlation with RT benefit (Figure 5). 

Although no significant difference was found regarding a RT Benefit in the ADT+ and ADT− patients (Mantel–Haenszel chi square 2-tailed *p*-value 0.26 or 0.99 for no benefit versus short remission or long-term over 12 month remission, respectively), we excluded patients with ADT from the analysis, since the success of salvage RT could not be clearly assessed for them. In this analysis, *MT-RNR1* transcript levels were clearly associated with remission and *IRF2* transcript levels with the recurrence of disease (Figure 6A,B, respectively). 

Additionally, we found six transcripts (*AEN*, *CDH1*, *IRF2*, *MYC*, *NFE2L2*, and *TCF3)* correlated with the Gleason score in the linear trend analysis (Figure 7) and eight transcripts (*CDKN1A*, *CLU*, *CCND1, MT-RNR1*, *CDK1*, *CDK6*, *ELK1*, and *STAT3*) correlated with TNM (Figure 8); a further four transcripts (*IGF1R*, *MAX*, *RAC1*, and *CD4*) had significant differences between TNM2 and TNM3, but overall the trend was not significant (Figure A3). 

Moreover, one transcript (*ALAS1*) was found to correlate with iPSA, two transcripts (*AKT1* and *CLU*) were found to correlate with PSA value at PSMA-PET, and three (*CDC25C*, *CTNNB1,* and *DDB2*) with peak uptake (Figure 9).

## 4. Discussion

Treatment concepts in oncology are becoming more individualized, which necessitates the use of valid biomarkers for decision making. The biomarker used primarily to monitor any therapy response in prostate cancer is still PSA, which does not fully reflect the tumor burden, metastatic stage, or possible treatment benefit or failure. Molecular marker analysis carries prognostic performance to the next level. The importance of tumors’ genetic profiling for the prognosis of prostate cancer in treatment decisions or in monitoring therapy responses has already been demonstrated for primary therapy or for the salvage radiotherapy of the former prostate bed and for metastatic disease [4,5,24,25,26,27,28,29,30,31]. Especially in the metastatic stage of prostate cancer, the interest in using liquid biopsies such as CTCs or nucleic acids from peripheral blood is accumulating since the histology of small metastases is not always available due to a lack of tissue material, and because blood sampling can be easily performed with minimal intervention and invasion. Thus, liquid biopsies represent an attractive tool and a good alternative to tissue-based molecular profiling.

In the present study, we evaluated the usefulness of CTC-derived RNA and plasma-derived RNA for the molecular analysis of a pre-defined gene panel and its possible predictive performance for RT benefit in oligometastatic prostate cancer relapses. Although we have a heterogeneous patient cohort, as it is generally in clinical praxis, a number of the investigated transcripts—relative to other established clinical or PSMA-PET-based parameters—showed a superior predictive potential to identify patients with a better response to salvage RT of metastases.

The overall dropout rate for CTC-derived RNA in our study was under 3% since CTCs were undetectable in only one blood sample. We used a specific EpCAM-positive CTC population (as recommended by the FDA, U.S. Food and Drug Administration) and were successfully analyzed the occurrence of the AR splice variants, some of which may predict treatment strategies and/or affect the response to therapy. We found multiple correlations and associations of the investigated AR variants with each other and with histopathological parameters, such as the TNM score, the Gleason score, and the number of active loci or tumor relapses, which represent tumor features and are linked to the aggressiveness of the disease. Our results are in line with published studies focused on investigating the correlative value of ARV expression as clinical biomarkers with disease dynamics and progression [18,32,33,34,35,36]. Of note, we found variability in the estimation of *AR-V567es* in the CTCs from prostate cancer samples in the three different detection systems used in this study, similar to previous studies [20]. Therefore, any conclusions drawn from the correlations between *AR-V567es* expression and clinical responses should be treated with caution, since they may not fully reflect the biological situation.

One recent study demonstrates that beyond ARVs, a distinct transcriptional profile in the CTCs from liquid biopsies can serve as an independent prognostic marker in patients with metastatic prostate cancer [37]. Despite such promising data, CTCs are not yet accepted in clinical use. One possible reason may be the challenge of isolating CTCs efficiently and another that their prediction accuracy is potentially hampered by intra-patient tumor heterogeneity. Another potential method for screening for possible biomarkers in metastatic diseases is the use of blood-derived nucleic acids. Several studies have reported the detectability of potential therapy-response biomarkers in whole blood mRNA or cfRNA (cell-free RNA) from plasma [34,35,38]. The isolation of blood-derived nucleic acids is relatively easy compared to CTCs; however, this source does not discriminate well between “tumor” and “normal” cell-derived nucleic acids, mostly representing the intrinsic genomic pattern of patients, which is also very important, especially for exploring the therapy response. Thus, a simultaneous examination of multiple parameters or components in liquid biopsies may facilitate a broader understanding from which to construct a more comprehensive molecular profile.

High-throughput technologies and bioinformatics tools, widely used in screening biomarkers for cancer diagnosis, treatment, and prognosis, have played an important role in recent years in developing prognostic classifiers for the prediction of clinical outcomes in prostate cancer patients. Based on clinical features, genetics, transcripts’ expression, or epigenetic profiles in tumors, the tumor microenvironment, or blood-derived nucleic acids or CTCs, a number of useful biomarkers have been reported, but most published studies have focused on the prediction of the pathogenesis and prognosis of the disease [35,36,37,38,39,40,41,42,43,44,45,46,47]. With regard to the RT outcome, either clinical parameters or gene expression profiles in tumors, xenografts, or cell lines have been investigated with respect to predicting the sensitivity of prostate cancer to RT [31,48,49,50]. To the best of our knowledge, our study is the first to explore the prognostic value of “liquid biopsies” for the benefit of salvage RT in oligometastatic prostate cancer relapses.

Here, we expanded the concept of liquid biopsy on blood-derived mRNA transcripts and demonstrated the clinical value of these parameters beyond the known associations with PSA, the Gleason score, and tumor categories for predicting RT benefit in our investigated cohort. An analysis of the entire cohort revealed two clinical parameters to correlate with RT benefit: the PSA (a known biomarker used to monitor therapy responses) value in PSMA-PET and the number of PSMA-positive lesions.

Transcriptional profile combinations or defined “transcriptional signatures” were found to correlate with the Gleason score or TNM score in our study. Identifying differentially expressed transcripts stratified by the Gleason or TNM scores provides valuable data on tumors’ potential aggressiveness. Using this approach, a number of previous investigations have constructed prognostic gene expression profiles, associated with the Gleason score, relapse, and prostate cancer mortality [51,52,53,54]. In our study, nine of the investigated transcripts built a “transcriptional signature” for the Gleason score (Figure A1) and two for the TNM score in the whole cohort (Figure A2). When the patients were stratified by their ADT statuses, five of the “Gleason score transcripts” (*AEN*, *CDH1*, *IRF2*, *MYC*, and *NFE2L2*) and one of the “TNM score transcripts” (*CDKN1A*) were retained in the ADT− patients in the correlation analysis (Figure 7 and Figure 8, respectively). The same transcript—*IRF2*—remained associated with disease recurrence, giving credibility to our results and some diagnostic potential.

The expression patterns of a number of transcripts were correlated with different clinical variables, such as iPSA, PSA doubling time, peak uptake, or the PSA value at PSMA-PET, which was also linked to their diagnostic values. In the current study, it was important to determine a gene expression pattern that could provide prognostic information for salvage RT in oligometastatic prostate cancer relapse patients beyond the clinical and pathologic features. Distinct “transcriptional signatures” were found for patients in short-term (<12 months) or long-term (≥12 months) remission (Figure 4 and Figure 5). These results were statistically significant in the correlation analysis, making those transcript combinations good prognostic candidates. Patients with such profiles have a significantly better chance for remission after salvage RT of the metastases.

When the patients for whom the success of RT could not be clearly assessed (due to concurrent ADT during follow-up) were excluded, *MT-RNR1* was found to be highly significantly correlated with RT benefit as the sole independent factor. An altered expression of this transcript had been previously observed in prostate cancer and colorectal tissues [55,56]. Of note, genetic variations in this region were found to act as prognostic markers for hepatocellular carcinoma and gastric cancer [57,58,59].

The results presented herein expanded the role of CTCs- or blood-derived RNAs and demonstrated their clinical value beyond cancer pathogenesis, providing benefits for diagnostic and/or prognostic biomarker development, as well as for improving therapeutic interventions. As examined in our study, multiple components of “liquid biopsy” provided a distinct molecular profile or “transcriptional signature” in the patients, who benefited from the salvage RT of prostate cancer metastases.

This study also has several limitations, namely, the number of patients enrolled in this study was limited, non-uniformly determined radiation doses were applied, and a number of patients received ADT in the follow-up; however, we were still able to demonstrate significant results.

## 5. Conclusions

In conclusion, our preliminary findings provide a pillar for exploring the prognostic value of liquid biopsies for the benefit of PSMA-PET-guided salvage radiotherapy in oligometastatic prostate cancer relapses. Further studies on larger and uniformly treated patient cohorts are necessary to validate our observations.

## Figures and Tables

**Figure 1 cancers-14-04095-f001:**
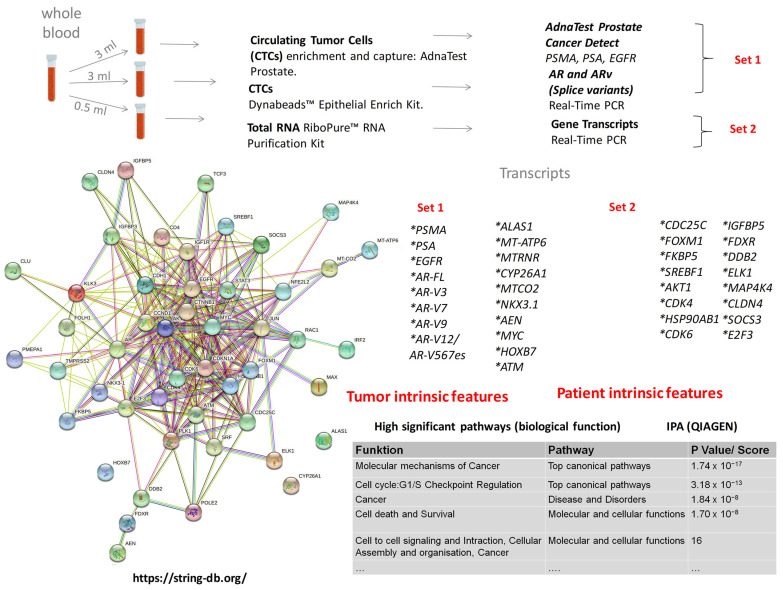
Study workflow. Processing patient blood samples (upper panel), candidate transcripts’ prioritization, and sets of investigated transcripts (lower panel).

**Figure 2 cancers-14-04095-f002:**
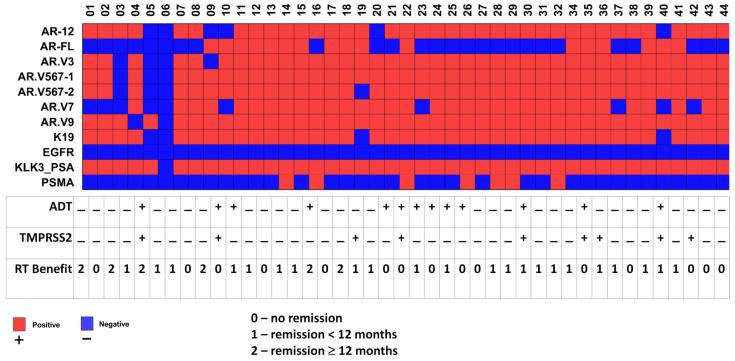
Landscape of investigated molecular alterations in individual CTCs and RT benefit. The overview of 43 samples with 42 CTC-positive samples in the investigated cohort is shown in the upper panel. The overall results of CTC analysis, ADT application in follow-up, and RT response are schematically summarized. The meaning of the color and number or “+/−“ coding is as indicated.

**Figure 3 cancers-14-04095-f003:**
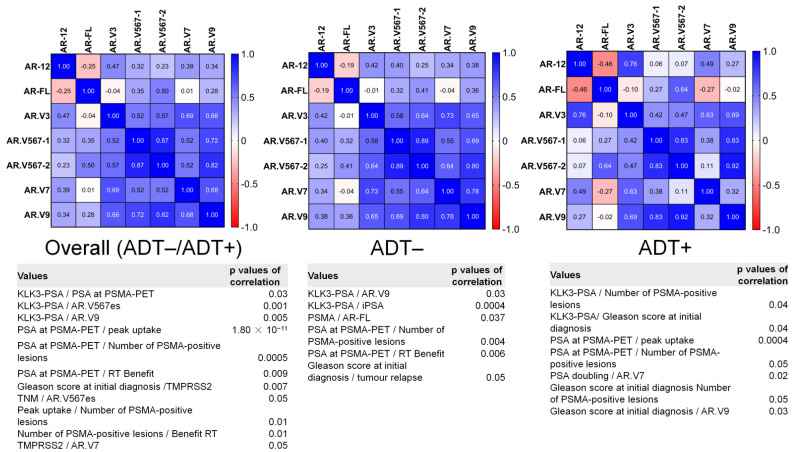
Correlation analysis. Heatmap profiles for the AR variant correlations in overall, ADT−, and ADT+ cohorts (upper panel). The overview of significant correlations for the clinical variables, stratified by ADT application or overall (lower panel). The meaning of the color-coding on the heatmaps is as indicated in the key, with 1 representing the highest positive correlation value and −1 representing highest negative correlation.

**Figure 4 cancers-14-04095-f004:**
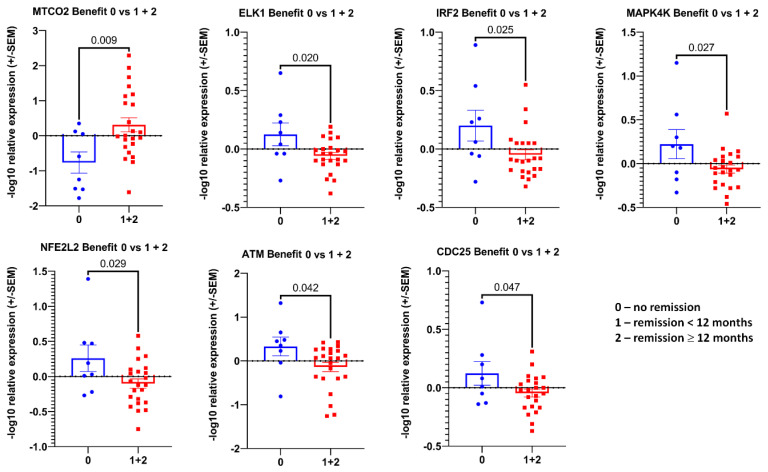
The association between transcripts’ expression in blood-derived mRNA and treatment response. Log10 relative expression levels of gene transcripts were tested for association with benefit. *p* values are indicated after unpaired t-tests between two groups. Only the significant associations in overall cohort are depicted for RT benefit. The meaning of the number-coding is as indicated in the figure key.

**Figure 5 cancers-14-04095-f005:**
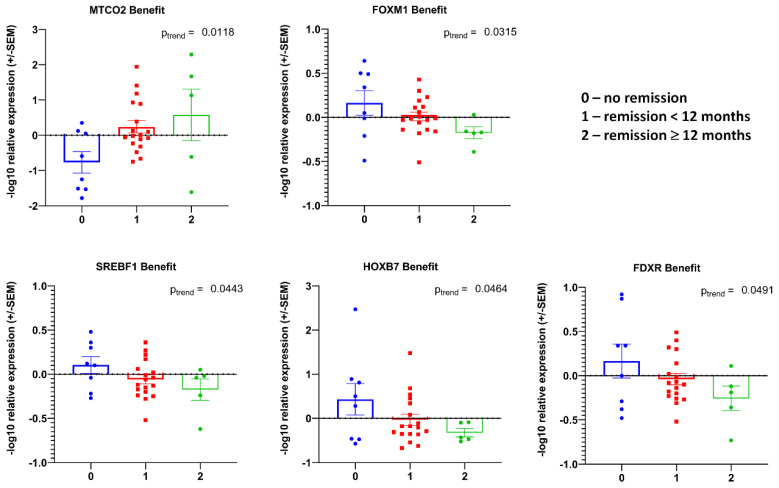
Trends in transcripts’ levels of expression from blood-derived mRNA and association with treatment response. Log10 relative expression levels of gene transcripts were tested for association with benefit. *p* values are indicated after linear test for trend following ANOVA. Only significant results in overall cohort are presented. The meaning of the number-coding is as indicated in the figure key.

**Figure 6 cancers-14-04095-f006:**
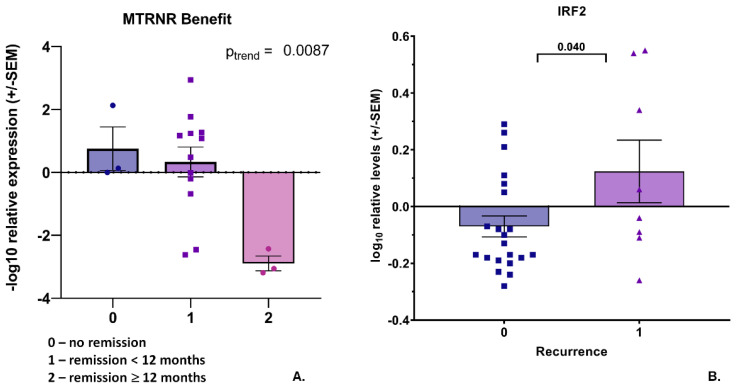
Associations between transcripts’ expression in blood-derived mRNA and treatment response or disease recurrence in ADT− cohort. (**A**) Bar graph showing −log10 relative expression levels of *MT-RNR1* in different benefit groups as specified in the key. *p* value indicated after linear test for trend following ANOVA. (**B**) Log10 relative expression levels of *IRF2* in groups with different disease recurrences. *p* value indicated after unpaired *t*-test between two groups.

**Figure 7 cancers-14-04095-f007:**
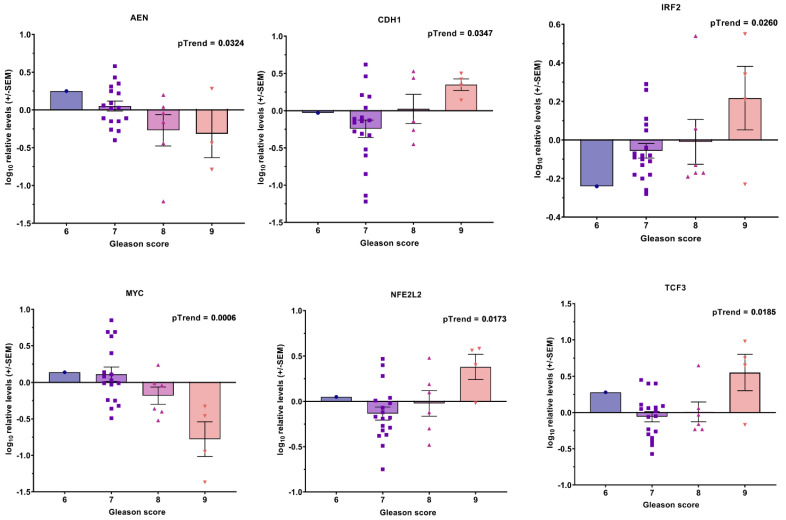
Associations between transcripts’ expression in blood-derived mRNA and Gleason score in ADT− cohort. Bar graphs showing −log10 relative expression levels of gene transcripts in different Gleason score groups. *p* values are indicated after linear test for trend following ANOVA.

**Figure 8 cancers-14-04095-f008:**
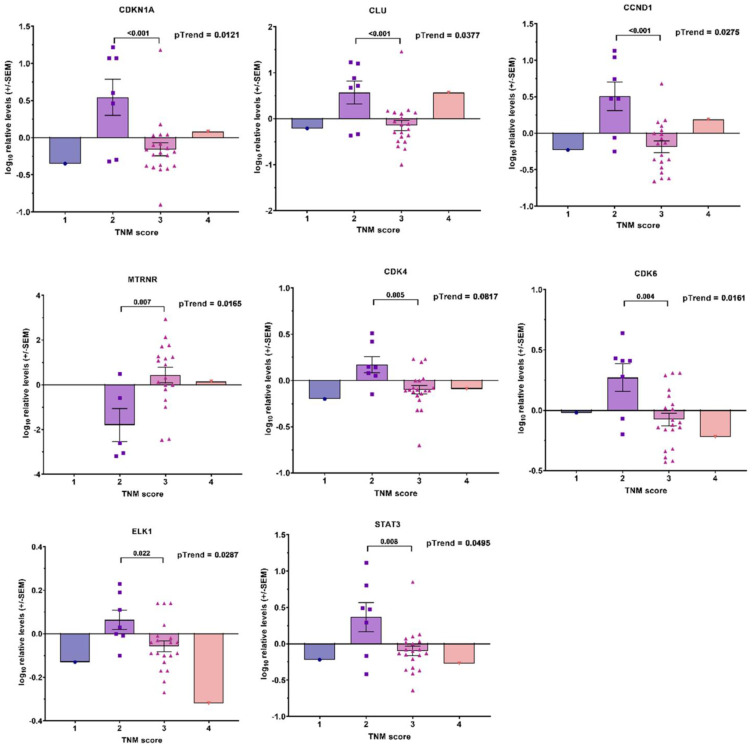
Associations between transcripts’ expression in blood-derived mRNA and TNM score in ADT− cohort. Bar graphs showing −log10 relative expression levels of gene transcripts in different TNM score groups. *p* values are indicated after linear test for trend following ANOVA or after unpaired *t*-test between two groups (TNM = 2 and TNM = 3).

**Figure 9 cancers-14-04095-f009:**
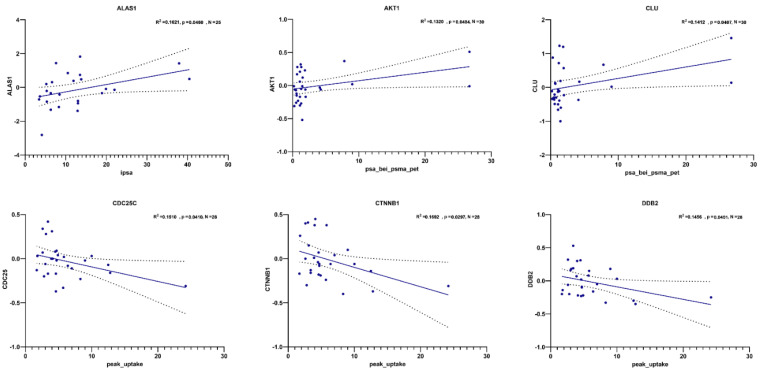
Correlations between transcripts’ expression in blood-derived mRNA and PSA-related standard clinicopathologic variables in ADT− cohort. Scatter plots showing correlation between −log10 relative expression levels of gene transcripts (on the *y* -axis) and different clinical parameters (on the *x*-axis). Pearson correlation R2 values are indicated, along with *p* values and N (number of observations/patients). In each graph, linear regression is denoted by the dark blue line and error bars are indicated by the black dotted lines.

**Table 1 cancers-14-04095-t001:** Patients’ characteristics.

Total Number of Patients/Blood Samples Available	44/43
**Age [years, Mean (Range)]**	63.6 (47–80)
**TNM at primary diagnosis [number of patients]**	
T1	6
T2	9
T3	27
T4	1
N 0	38
N 1	5
M 0	43
M 1	0
**Gleason—Score at primary diagnosis [number of patients]**	
Gleason sum 6	1
Gleason sum 7a	22
Gleason sum 7b	3
Gleason sum 8	7
Gleason sum 9	10
Gleason sum 10	0
**Therapy after primary diagnosis**	
Prostatectomy	39
Radiotherapy	4
**PSA Level in Blood sample [ng/mL, mean (range)]**	
At primary diagnosis	12.15 (2.6–40.4)
At relapse and PSMA-PET	5.15 (0.19–64)
Doubling time at PSMA-detected relapse [month]	22 (1–46)
**Site of relapse [number of patients]**	
Former prostate region only	3
Pelvic lymphatic nodes only	22
Extra pelvic lymphatic nodes only	0
Bone metastases only	7
Metastases at lung/liver only	1
Both pelvic and extra pelvic lymphatic nodes	2
Both lymphatic nodes and bone metastases	4
Both former prostate region and bone metastases	2
Both former prostate region and pelvic lymphatic nodes	2
**Peak uptake [mean (range)]**	7.7 (1.8–49.8)
**Number of patients not receiving ADT during follow up (ADT−)**	30
**Number of patients receiving ADT during follow up (ADT+)**	13
**Treatment outcome after Radiotherapy of relapse**	
Time to progression [month, mean (range)]	7.1 (2–19)
**ADT− patients**	
Remission < 12 months [number of patients]	20
Remission ≥ 12 months [number of patients]	3
No remission	7
**ADT+ patients**	
Remission < 12 months [number of patients]	5
Remission ≥ 12 months [number of patients]	2
No remission	6

## Data Availability

The data presented in this study are available upon reasonable request from the first or corresponding author.

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
