# Peer review of "The Prognostic Value of Liquid Biopsies for Benefit of Salvage Radiotherapy in Relapsed Oligometastatic Prostate Cancer"

_cancers, 2022, doi:10.3390/cancers14174095_

Round 1

Reviewer 1 Report

A very detailed and sound study on the role of liquid biopsies in prostate cancer with the focus on the very important clinical issue of salvage radiation therapy in metastatic disease. The authors correctly indicate the limitations of this study but can still provide some significant data that gives this paper hifg significance and pave the way for further investigations.

The influence of genetic factors on the prognosis of prostate cancer has been known for years. The authors have investigated whether this genetic influence can also be detected on the prognosis after radiotherapy in oligometastasized prostate cancer, diagnosed in the PSMA-PET-CT. The results are highly relevant because radiotherapy of oligometastasized prostate cancer is a new method whose significance is only gradually being clarified. The authors were able to demonstrate that specific genes and their methylomas are of significant importance for the prognosis after radiotherapy. The genetic influence is very large. For this reason, the authors were also able to demonstrate significant influences in their small group of patients examined. A larger group of patients could also help identify genes of lesser influence. A longer follow-up period could also identify genes whose effect affects slow-growing tumors. Subsequent publications are promising. The conclusions for the genes examined are methodologically clearly justified and provide a clear answer to the question explained in the introduction. The cited references are extensive. They contain in particular the basic publications of Den RB 2014 and the subsequent current publications. The tables are very extensive and will only interest those readers who want to carry out their own investigations or compare their own results.

Author Response

Dear reviewer,

I sincerely thank you for your careful review of our manuscript and for your helpful advice. We have implemented your advice and made small corrections.

best regards

Roland Merten

Reviewer 2 Report

This study was reported the prognostic value of liquid biopsies in patients with prostate cancer who had oligometastasis and relapse after definitive therapy. Overall, this paper is well written. The reviewer would like to suggest some critiques as follows.

1.     The title is unclear. “in relapsed oligometastatic prostate cancer” may be better.

2.     On line 99, this is a grammatical error.

3.     On line 109, the authors described “completed definitive RT.” Haw many grays of radiation did the patients received?

Author Response

(The authors gave the same response as above.)
